# Improved Environment Stability of Y_2_O_3_ RRAM Devices with Au Passivated Ag Top Electrodes

**DOI:** 10.3390/ma15196859

**Published:** 2022-10-02

**Authors:** Hae-In Kim, Taehun Lee, Won-Yong Lee, Kyoungdu Kim, Jin-Hyuk Bae, In-Man Kang, Sin-Hyung Lee, Kwangeun Kim, Jaewon Jang

**Affiliations:** 1School of Electronic and Electrical Engineering, Kyungpook National University, Daegu 41566, Korea; 2School of Electronics Engineering, Kyungpook National University, Daegu 41566, Korea; 3School of Electronics and Information Engineering, Korea Aerospace University, Goyang 10540, Korea

**Keywords:** sol-gel, Y_2_O_3_, RRAM, environment stability, passivation

## Abstract

In this study, we fabricated sol-gel-processed Y_2_O_3_-based resistive random-access memory (RRAM) devices. The fabricated Y_2_O_3_ RRAM devices exhibited conventional bipolar RRAM device characteristics and did not require the forming process. The long-term stability of the RRAM devices was investigated. The Y_2_O_3_ RRAM devices with a 20 nm thick Ag top electrode showed an increase in the low resistance state (LRS) and high resistance state (HRS) and a decrease in the HRS/LRS ratio after 30 days owing to oxidation and corrosion of the Ag electrodes. However, Y_2_O_3_ RRAM devices with inert Au-passivated Ag electrodes showed a constant RRAM device performance after 30 days. The 150 nm-thick Au passivation layer successfully suppressed the oxidation and corrosion of the Ag electrode by minimizing the chance of contact between water or oxygen molecules and Ag electrodes. The Au/Ag/Y_2_O_3_/ITO RRAM devices exhibited more than 300 switching cycles with a decent resistive window (>10^3^). They maintained constant LRS and HRS resistances for up to 10^4^ s, without significant degradation of nonvolatile memory properties for 30 days while stored in air.

## 1. Introduction

Complementary metal oxide semiconductor (CMOS)-based data storage devices, such as flash drives, are considered and used as critical components in ubiquitous applications. The writing and erasing processes of flash memory involve storing electrons and removing the stored electrons inside the floating gate, respectively. One critical issue is that to maintain the stored electrons for nonvolatile memory properties, the tunneling oxide layer should be 8 nm, minimizing the loss of the stored electrons from the floating gate layer, leading to a scaling margin [1,2,3].

To solve these issues, resistive random-access memory (RRAM) has been used as a candidate for next-generation nonvolatile memory [4,5,6]. Typically, RRAMs have a simple metal-active layer-metal structure, exhibiting low power consumption, fast writing speed, and extreme scalability. Recently, to realize a neuromorphic computing system that mimics the human brain, RRAMs have been used because of the aforementioned advantages. Many metal oxides such as SiO_x_, ZrO_2_, TiO_x_, Hf_x_O, and Y_2_O_3_ have been studied for use in RRAM devices [7,8,9,10,11,12,13,14,15,16,17]. Among these, Y_2_O_3_ demonstrates advantages over other metal oxides. Y_2_O_3_ has a high dielectric constant, a large optical bandgap, and significant thermal stability. For these reasons, Y_2_O_3_ has been studied as a compromise for high-k insulators to replace the SiO_2_ dielectric layer in the CMOS process [18,19]. Additionally, the fast ion movement inside Y_2_O_3_, related to the write/erase operation speed of RRAM devices, is another advantage. Based on the unit RRAM devices, the sneak path problem should be solved to realize an RRAM array. Combining transistors as selectors is a well-known solution to this problem [20,21]. The combined transistor should have extreme bias stability. Recently, emerging metal oxide transistors with Y_2_O_3_ passivation layers have been shown to exhibit bias stability. Using Y_2_O_3_ layers as active channel layers for RRAM devices and as passivation layers for emerging metal oxide transistors simultaneously helps reduce the number of steps and costs required to realize a 1T-1R RRAM array [22,23]. 

In terms of low-cost fabrication, the sol-gel process is promising. High-quality pure metal oxide layers can be easily formed. Their structural, optical, chemical, and electrical properties can be tuned by changing the annealing conditions or the precursor component ratio [24,25,26,27]. The starting precursors for the sol-gel process are in the liquid phase. Liquid-phase precursors can be used for spin-coating, dip-coating, or printing techniques, allowing for large-area applications at a low cost. This eliminates the need to use conventional complex and expensive vacuum-based deposition methods to form high-quality pure metal oxides for electric devices, such as thin-film transistors, sensors, and RRAM devices [8,22,23,28,29]. 

In this study, RRAM devices consisting of sol-gel-processed Y_2_O_3_ active layers were fabricated on ITO/glass substrates. Ag was used as the top electrode and source for conductive filament formation inside the Y_2_O_3_ layers. The fabricated RRAM devices showed conventional bipolar operation. However, after one month, the typical device performance was noted to have degraded significantly owing to the surface oxidation of the top Ag electrodes. In contrast, a fabricated RRAM device with Au-passivated Ag top electrodes showed improved environmental stability and consistently promising RRAM device performance after one month. The fabricated RRAM devices showed a high resistance state (HRS)-to-low resistance state (LRS) ratio of ~10^3^, a SET voltage of +2.5 V, and a RESET voltage of −4.0 V. The RRAM devices also showed promising nonvolatile memory properties. 

The electrical characteristics of RRAM devices can be affected by the chemical reaction between electrodes and active channel layers [30,31]. The ambient conditions can also affect the electrical characteristics and the long-term environmental stability of RRAM devices. Unfortunately, only a few investigations were reported for oxygen vacancy-based RRAM devices [32]. Several investigations for electrochemical metallization RRAM devices showed the effect of humidity on device performance, boosting metal ion concentration due to the incorporation of moisture into electrochemical metallization RRAM devices [33,34,35,36]. This makes conductive metal-based filament form easily, decreasing SET voltages. However, our results showed a different trend than previous research results. Top electrode corrosion was dominant in degrading the RRAM device performance without a change in SET/RESET voltages. For the first time, the top electrode corrosion issue should be considered to confirm the long-term stability of electrochemical metallization RRAM devices and suggest a simple strategy with thick inert Au passivation layers.

## 2. Materials and Methods

In this experiment, two types of RRAM devices were manufactured to investigate the passivation effect of the Au layer on the Ag top electrode: Ag/Y_2_O_3_/ITO/glass and Au/Ag/Y_2_O_3_/ITO/glass structures. The Y_2_O_3_ precursor was prepared by dissolving 0.3 mol of yttrium (III) nitrate tetrahydrate (Y(NO_3_)_3_·4H_2_O; Sigma Aldrich, St. Louis, MO, USA, 99.9%) in 5 mL of 2-methoxyethanol (Sigma Aldrich, anhydrous, St. Louis, MO, USA, 99.8%). To ensure the uniformity of the solution and prepare a clear solution, sonication was performed for 10 min at room temperature. ITO-coated glass was used as the substrate (Sigma Aldrich, surface resistivity 70–100 ohm/sq, St. Louis, MO, USA, slide). The substrates were cleaned with acetone and deionized water using a sonicator for 10 min each. To remove the remaining organic contaminants, the ultraviolet/ozone process was performed for 1 h. The prepared Y_2_O_3_ precursor was then deposited on the cleaned substrate through a spin-coating process at 3000 rpm for 50 s. The remaining solvent was removed using a hot plate at 100 °C for 10 min. An additional annealing process was performed at 500 °C for 3 h in a furnace to convert them into Y_2_O_3_ films. Subsequently, a 20 nm Ag top electrode was deposited using the thermal evaporation method at 5.5 × 10^−6^ Torr and a deposition rate of 1.5 Å/s. For the Au-passivated electrode, an additional 100 nm Au layer was deposited on top of the 20 nm thick Ag top electrodes. All metal layers were patterned during the deposition process using a metal shadow mask of size 30 μm × 30 μm. (Figure 1)

The crystal structure and phase were analyzed using grazing incidence X-ray diffraction (GIXRD, X’pert Pro, Malvern PANalytical, Malvern, UK; incident angle = 0.3°, Cu-Kα, λ = 1.54 Å). X-ray photoelectron spectroscopy (XPS; Nexsa, Thermo Fisher, MA, USA) was used to investigate the chemical composition of the fabricated RRAM devices. A monochromatic AlKα (1486.6 eV) source was used. The spot size was 400 um. The high-resolution scans were obtained with the pass energy of 50 eV and 0.1 eV per step. All measurements were carried out at 1.0 × 10^−7^ Torr. Origin software was used to perform peak-fitting and compositional analysis. The electrical characteristics were examined using a probe station (MST T-40000A, Hwaseong, Korea) equipped with a KEITHLEY 2636B Source Meter (Keithley Instruments, Cleveland, OH, USA) at room temperature.

## 3. Results and Discussion

Figure 2 shows the GIXRD spectra of the Ag/Y_2_O_3_/ITO films on glass substrates. Peaks related to Ag, Y_2_O_3_, and ITO were observed, confirming that all layers have a polycrystalline structure. The peaks at 2θ = 38.26°, 44.47°, and 64.71° correspond to the (111), (200), and (220) planes of cubic Ag, respectively (JCPDS No. 98-0719). The deposited Y_2_O_3_ films showed a cubic structure, which is stable at low temperatures. The peak at 2θ = 29.15° represented the (222) plane of the cubic structure of Y_2_O_3_ (JCPDS No. 05-0574). The next peak at 2θ = 30.54° corresponded to the (222) plane of the cubic structure of ITO (JCPDS No. 89-4598), respectively. The crystalline sizes in each layer were calculated using the Scherrer equation:D = (0.9 λ)/(β Cos θ)(1)
where D, λ, β, and θ indicate the crystalline size, X-ray wavelength (1.54 Å), line broadening at half the maximum intensity, and Bragg angle, respectively. Crystalline sizes calculated for the Y_2_O_3_, ITO, and Ag layers were 8.84, 9.66, and 11.72 nm, respectively.

The chemical compositions of the films were analyzed using the XPS data. The collected data were calibrated based on the position of the C1s transition line at 284.8 eV. Figure 3a–c shows the Ag 3d XPS data of Ag and Au-passivated Ag electrodes, respectively. Clearly intensified signals corresponded to AgO and AgO_2_ after being stored for one month in air. In contrast, in Au passivated Ag electrodes, the peak at 368.3 eV corresponding to Ag was dominant. Figure 3d show the Au 4f doublet XPS spectral region of Au passivation layers. Figure 3e,f show the O1s XPS spectra of the Ag and Au-passivated Ag electrodes, respectively. The peak at 530.1 eV corresponded to Ag_2_O, while the peak at 531.0 eV corresponded to AgO [37,38,39]. The peaks at 532.0 and 533.0 eV corresponded to absorbed OH groups and structural water molecules, respectively [40]. However, GIXRD showed no AgO or Ag_2_O-related peaks, indicating an amorphous phase.

In contrast, the Au-passivated Ag films, after being stored for one month in air, did not show any related peak in Figure 3b. This indicates that the top Ag electrode can easily oxidize due to air-exposure-induced oxidation and contamination. However, a continuously deposited Au passivation layer on top of Ag electrodes can successfully suppress the air-exposure-induced oxidation process and minimize contamination. 

Figure 4 shows representative I–V curves of the fabricated RRAM devices. All the fabricated devices exhibited conventional bipolar RRAM switching characteristics and did not require the forming process, regardless of the additional deposition of the passivation layer. To observe the RRAM characteristics, a sweeping voltage was applied in the range of −10.0 to 5.0 V. The fabricated RRAM device was initially in a high resistance state (HRS). During the positive voltage sweep, the current abruptly increased at ~ +2.5 V, indicating that the device had attained a low resistance state (LRS). The voltage at which the RRAM device status changes from HRS to LRS is referred to as the SET voltage. In contrast, during the negative voltage sweep, the current abruptly decreased at –5.5 V, indicating that the device had returned to the HRS. The voltage at which the RRAM device status changes from LRS to HRS is referred to as the RESET voltage. 

In Y_2_O_3_-based RRAM, such resistance switching memory behavior can occur based on both oxygen vacancies (OxRRAM) and the migration of metal ions (CBRAM). However, in our previous study, we demonstrated that the Ag/Y_2_O_3_/ITO device is a type of CBRAM by showing that the Au/Y_2_O_3_/ITO device, which uses an inert material as a top electrode, does not exhibit RRAM characteristics [15]. In addition, the fabricated RRAM device did not require an initial forming process. It is well known that the active layer in defect-rich materials or RRAM devices, consisting of Ag or Cu, does not require an initial forming process [41,42]. As with the general characteristics of CBRAM, the Ag/Y_2_O_3_/ITO RRAM device operates via the migration of Ag ions and subsequent reduction and oxidation reactions. When the Ag top electrodes are biased to a positive voltage sweep, the oxidation process forms Ag ions at the Ag/Y_2_O_3_ interface. The formed Ag^+^ ions drift to the negatively biased ITO bottom electrode and reduce to Ag atoms that receive electrons from the electrode at the Y_2_O_3_/ITO interface, initiating the formation of Ag conductive filaments. As a result, conductive paths consisting of Ag atoms are formed from the Ag top electrode to the ITO bottom electrode, enabling a large current flow. This is called the SET process, in which the device enters the LRS. When a negative voltage is applied to the top electrode, the oxidized Ag atoms in the conductive paths return to the top electrode. As a result, the Ag conductive paths are ruptured, limiting the current flow. This is called the RESET process, and the device returns to the HRS. 

After storage for one month in air, the representative device parameters, such as SET, RESET, HRS, and LRS were extracted. The statistical data of the extracted SET and RESET voltages and HRS and LRS values are plotted in Figure 5. The HRS and LRS values were calculated with resistance values at +0.1 V of the read voltage. After storage for one month in air, the SET, RESET, and HRS values did not change significantly. However, a significant increase in the LRS values for Ag/Y_2_O_3_/ITO RRAM devices was observed. Such degradation of the LRS value was not observed in the Au/Ag/Y_2_O_3_/ITO device after storage for one month in air.

It is well known that water molecules from the environment or surface oxidation of electrodes can affect RRAM device performance and lead to the degradation of long-term environmental device stability. For example, the water molecules can be easily absorbed on the Y_2_O_3_ layers through nano-porous top Ag electrodes. This can increase hydroxide ion (OH^−^) concentration at the Y_2_O_3_/ITO interfaces. In addition, the water molecules can also be absorbed on the grain boundary surface of the Y_2_O_3_ layers. It is well known that to render the Y_2_O_3_ films electrically neutral, the Ag+ concentration will change based on the quantity of hydroxide. The Ag^+^ concentration was increased as a hydroxide concentration was increased. The increased Ag^+^ concentration in Y_2_O_3_ led to easier conductive filament formation, resulting in lower SET voltages [33,34,35,36]. In addition, it is easy to break the cond/uctive filament, leading to a reduced size of the broken gap and thereby resulting in lower HRS values. However, in this experiment, after storage for one month in air, the SET, RESET, and HRS values did not change significantly. These parameters originate from strong hydrogen bonds and do not change easily under environmental conditions. LRS values increased significantly. For these reasons, even though –OH groups related peaks are observed in the XPS data in Figure 3e, water molecules from the environment were not the main critical factors for degrading the long-term stability of the RRAM devices. The oxidized Ag electrodes are a more critical factor for the degradation of the LRS values in the XPS data [43]. After exposure to air for one month, the XPS results displayed peaks corresponding to AgO and Ag_2_O. They may be formed from pure Ag top electrodes, leading to an increase in resistance after one month of storage in air. This leads to an increase in LRS. The fabricated RRAM device with inert Au-passivated Ag electrodes did not show any Ag_2_O, absorbed oxygen, or absorbed OH groups based on the XPS data. Figure 6 shows the schematics of the potential mechanism of water or oxygen molecule effects on oxidation of Ag electrodes and Au passivation effect, respectively.

The nonvolatile memory properties of the fabricated devices were evaluated on day 30 by estimating the endurance and retention characteristics (Figure 7). The Ag/Y_2_O_3_/ITO RRAM devices exhibited poor endurance properties. They also exhibited considerably degraded endurance characteristics of approximately 50 switching cycles with a relatively small resistive window (>10). In addition, the LRS resistance increased abruptly near 10^2^ s in the retention measurements. The HRS resistance of the Ag/Y_2_O_3_/ITO RRAM device is relatively high. The active top Ag electrodes are highly prone to oxidation and corrosion under the application of a positive bias when the electrode is exposed to water molecules. This oxidation and corrosion process was accelerated during the constant positive bias step of the endurance test, leading to poor endurance performance. In contrast, the Au/Ag/Y_2_O_3_/ITO RRAM device exhibited good memory characteristics over time. The Au/Ag/Y_2_O_3_/ITO RRAM devices exhibited better endurance properties under a constant positive bias. The relatively thick inert Au layers minimized the aforementioned oxidation and corrosion process originating from water or oxygen molecules from the environment. The Au/Ag/Y_2_O_3_/ITO RRAM devices exhibited more than 300 switching cycles with a decent resistive window (>10^3^). They maintained constant LRS and HRS resistances up to 10^4^ s without significant nonvolatile memory properties. Table 1 compares the resistive switching characteristics of Y_2_O_3_ RRAM devices [44,45,46]. The fabricated sol-gel processed RRAM devices, not using conventional deposition techniques, showed a larger HRS/LRS ratio. For the first time, the top electrode corrosion issue should be considered to confirm the long-term stability and suggest a simple strategy with thick inert Au passivation layers. The Y_2_O_3_ RRAM device with Au passivated Ag top electrodes showed long environmental stability.

## 4. Conclusions

The Sol-gel-processed Y_2_O_3_-based RRAM devices were fabricated. The fabricated Y_2_O_3_ RRAM devices exhibited conventional bipolar RRAM device characteristics, which do not require the forming process. The long-term stability of the RRAM devices was investigated. The Y_2_O_3_ RRAM devices with a 20 nm thick Ag top electrode showed an increase in the LRS and HRS and a decrease in the HRS/LRS ratio after 30 days owing to oxidation and corrosion of the Ag electrodes. However, Y_2_O_3_ RRAM devices with inert Au-passivated Ag electrodes showed a constant RRAM device performance after 30 days. Top electrode corrosion was a dominant factor in degrading the RRAM device performance without a change in SET/RESET voltages. For the first time, the top electrode corrosion issue should be considered to confirm the long-term stability of electrochemical metallization RRAM devices and suggest a simple strategy with thick inert Au passivation layers. The 150 nm-thick Au passivation layer successfully suppressed the oxidation and corrosion of the Ag electrode by minimizing the chance of contact between water or oxygen molecules and Ag electrodes. This method is a simple yet promising tactic to confirm the long-term stability of next-generation RRAM devices with Ag and Cu electrodes.

## Figures and Tables

**Figure 1 materials-15-06859-f001:**
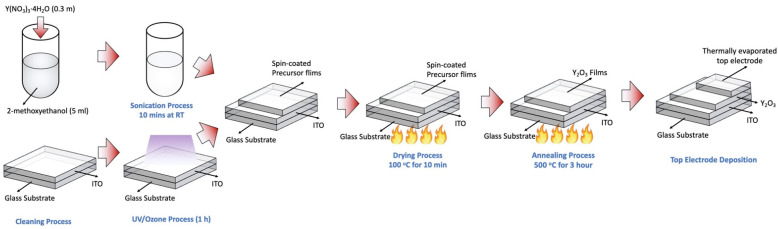
Fabrication process of sol-gel processed Y_2_O_3_ RRAM devices.

**Figure 2 materials-15-06859-f002:**
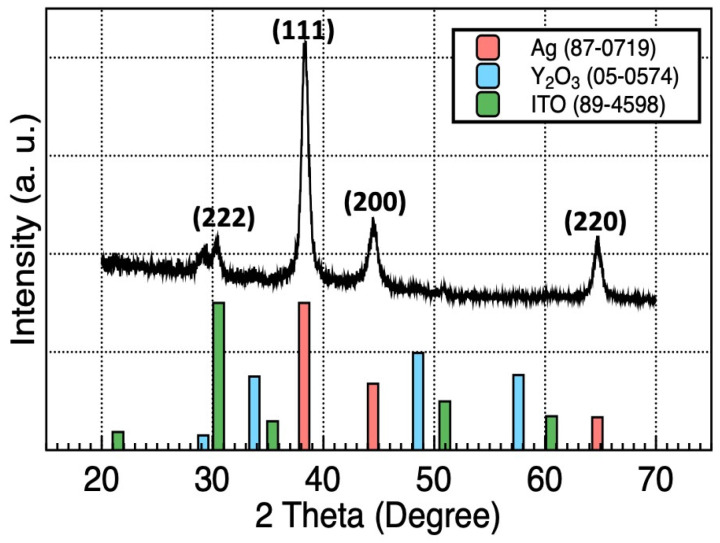
GIXRD spectra of Ag/Y_2_O_3_/ITO/Glass films.

**Figure 3 materials-15-06859-f003:**
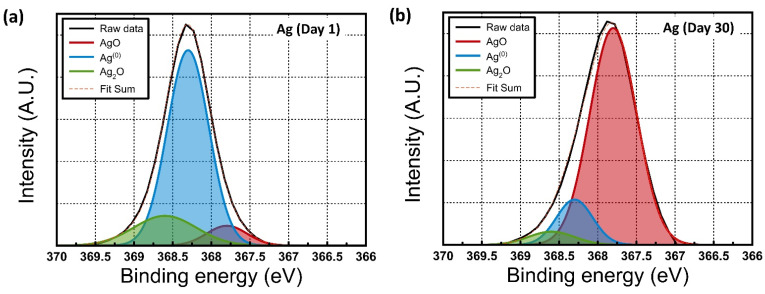
**(a**–**c**) Ag 3d XPS data for Ag and Au-passivated Ag electrodes. (**d**) Au 4f doublet XPS spectral region of Au passivation layers. (**e**,**f**) O1s XPS data for Ag electrode and Au-passivated Ag electrodes, respectively, after being stored for one month in air.

**Figure 4 materials-15-06859-f004:**
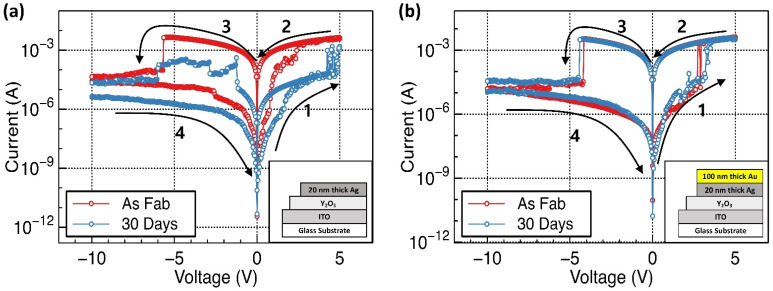
The representative I–V curves of pristine and 30-day stored Y_2_O_3_ RRAM devices: (**a**) with Ag top electrodes and (**b**) Au-passivated Ag top electrodes, respectively. The insets show the schematic images of fabricated RRAM devices.

**Figure 5 materials-15-06859-f005:**
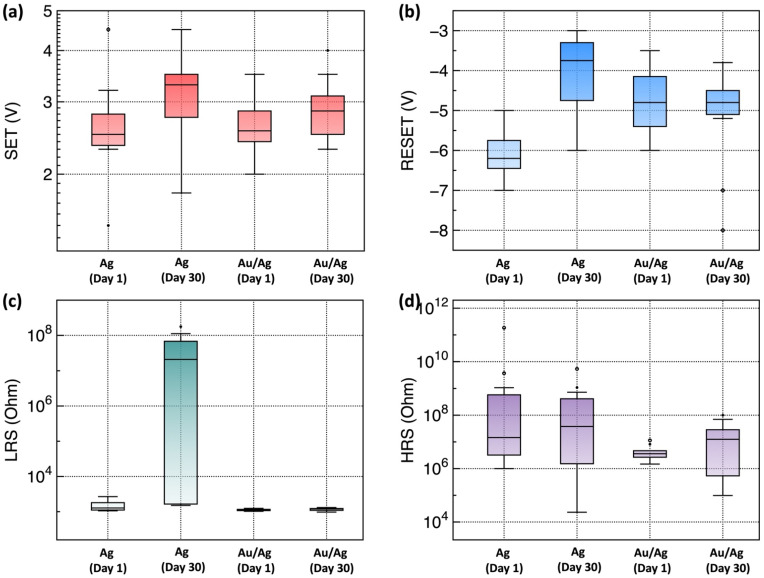
Extracted performance parameters of the fabricated Y_2_O_3_ RRAM devices with Ag and Au-passivated Ag top electrodes: (**a**) SET voltage, (**b**) RESET voltage, (**c**) LRS, and (**d**) HRS. The dots are outliners which are located outside the whiskers of the box plot.

**Figure 6 materials-15-06859-f006:**
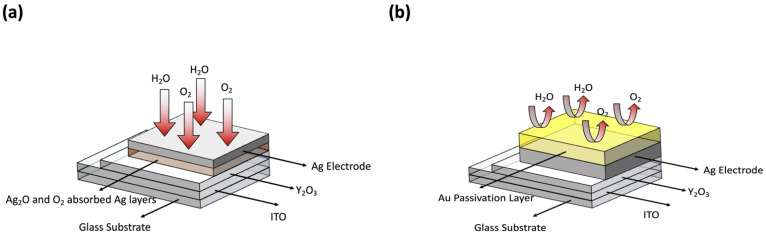
The schematics of the potential mechanism of water or oxygen molecules effects on oxidation of Ag electrodes (**a**) and Au passivation effect (**b**), respectively.

**Figure 7 materials-15-06859-f007:**
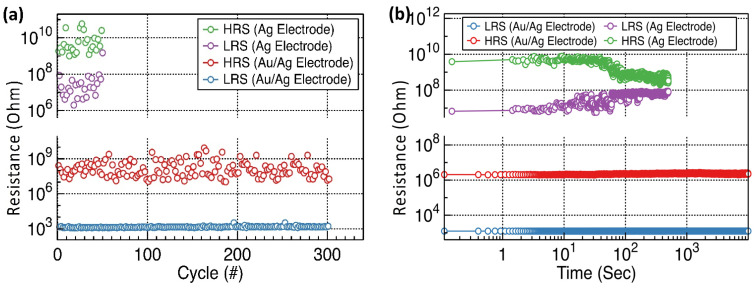
The nonvolatile memory characteristics of 30-day stored Y_2_O_3_ RRAM devices: (**a**) Representative endurance characteristics of Y_2_O_3_ RRAM devices with Ag and Au/Ag electrodes and (**b**) representative retention characteristics of Y_2_O_3_ RRAM devices with Ag and Au/Ag electrodes.

**Table 1 materials-15-06859-t001:** Comparison of Y_2_O_3_ RRAM device performance parameters.

Reference	Material System	Process	HRS/LRS	Endurance (cycle) /Retention (Sec)	Environment Stability
44	Al/Y_2_O_3_/Al	Ion beam sputter	~30	~3 × 10^4^/10^5^	N/A
45	n-Si/a-Y_2_O_3_/Y_2_O_3_/Al	Ion beam sputter	~10	~3 × 10^4^/~10^3^	N/A
46	Ni/Y_2_O_3_ /TiO_x_/TaN	E-beam Evap.	~10^2^	~10^2^/~10^4^	N/A
This Work	Au/Ag /Y_2_O_3_/ITO	Sol-gel	~10^3^	~3 × 10^2^/~10^4^	30 Days

## Data Availability

Data are available in a publicly accessible repository.

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
