# Peer review of "Improved Environment Stability of Y_2_O_3_ RRAM Devices with Au Passivated Ag Top Electrodes"

_materials, 2022, doi:10.3390/ma15196859_

Round 1
Reviewer 1 Report
In this paper, the authors fabricated the Y2O3-based RRAM devices by a sol-gel process. The precursor was mixed and spin-coated on the substrate, and two kinds of electrodes were deposited for comparison of the test stability. The paper and structure of the paper are well constructed, but the paper lacks novelty, and the reports of the devices do not present improvement in the field of either science or applications, therefore, I suggest rejecting the paper unless more solid evidence is provided.
The reported structure of the device is very simple and does not show enhancement compared with other reported papers. The authors should not just compare the two devices they fabricated.
In addition, although they proposed simple methods by spin-coating and sol-gel, the quality and thickness of the oxide layer are very hard to control for the real fabrication in industry, thus it is hard to control the quality of the products.
Reviewer 2 Report
The authors have presented a detailed study of Y2O3 based RRAM devices and the findings of environmental stability of these devices could be useful for researchers working in this field. However, several important control experiments are missing and the claims of this article remain unsubstantiated without additional data. Hence, I recommend this paper be rejected and reconsidered if the concerns/suggestions below are addressed.
Page 2, Line 47 - add a short description of sneak path problem and how explain how the Y2O3 based RRAM devices can be solution to the problem
Page 2, Line 85 - Add UV/Ozone process conditions (substrate temperature, tool description, plasma power) for the user to be able to replicate the cleaning process.
Page 3, Line 45 - Add more details for XPS measurements - spot size, scanning speed, resolution of the instrument, X-ray used, operating pressure, background correction and peak fitting method used etc.
Page 4, Figure 2 -
-The XPS spectra shows two peaks for OH, one for adsorbed and another related to structural water molecules. No explanation is provided as to what the difference between the two peaks is. The broader - OH peak is likely a background correction and the authors should be careful when fitting XPS spectra.
- Add Ag3d and Au 3d peaks in supplemental information to validate the theory of surface oxidation of Ag thin films.
- Also, add XPS data for Bothe Ag and Ag/Au electrodes from day 1 and day 30 of air exposure for comparison with Figure 4 (electronic properties).
Also explain/comment why GIXRD data does not show any Ag2O peaks.
Page 6, Line 79 - Explain or elaborate how Ag+ concentration was increased?
Page 6, Line 206 - Increase in HRS and LRS values is reported here but it contradicts with the data shown in Figure 4 where the HRS for the RRAM device with Ag top electrode remains unchanged (or not does not significantly change) with 30 days of air exposure while the LRS values increases.
Page 5, Line 176 - Data in Figure 4 contradicts with this statement, the LRS values after 30 days of air exposure increases while the statement suggests the opposite.
Page 7, Figure 5b - Show the retention characteristics over time for both the RRAM devices -with Au/Ag electrodes and Ag electrodes. That data is key to establish the main claim that this study makes regarding the improved RRAM properties with the Au/Ag electrodes when compared to the Ag electrodes.
Thanks

Reviewer 3 Report
Dear Editor,
I have read the manuscript entitled: “Improved Environment Stability of Y2O3 RRAM Devices with Au Passivated Ag Top Electrodes” and I would like to address following suggestions to the authors:
1. Please add some lines to indicate the novelty of your study, compare the results with that of the literature and emphasize the novelty of this study.
2. For each research method, it is necessary to expand the discussion. Please add schematic diagram for this study (preparation by sol-gel method).
3. Please describe all devices used in this paper (company, …).
4. Please explain more about sol-gel method in introduction and use related papers such as:
Sabbagh, F., K. Kiarostami, N.M. Khatir, S. Rezania, I.I. Muhamad, and F. Hosseini. Polymer Testing, 93, 106922, (2021).
Sabbagh, F., K. Kiarostami, N. Mahmoudi Khatir, S. Rezania, and I.I. Muhamad. Polymers, 12, 861, (2020).
Khatir, N.M., Z. Abdul-Malek, A.K. Zak, A. Akbari, and F. Sabbagh. Journal of Sol-Gel Science and Technology, 78, 91-98, (2016).
Khatir, N.M. and F. Sabbagh. Materials, 15, 5536, (2022).
In the conclusion, the performance findings of the research should have been summarized the innovations and future scope of the work should be highlighted more.

Round 2
Reviewer 1 Report
The authors need to compare their devices with other reports to emphasize the importance of their work and how good the results are.
Reviewer 2 Report
The authors have addressed most of the concerns except the most important one being that the claim that authors make for Au/Ag electrodes being better than Ag electrodes needs to be shown by showing improved RRAM properties. If the authors can't show this and show this performance repeatably, then this study does not add value to the scientific field.
Reviewer 3 Report
Accept in present form
Author Response
N/A